# Hierarchical rupture growth evidenced by the initial seismic waveforms

Takashi Okuda[1] & Satoshi Ide[1]

The ability to predict the eventual size of an earthquake during its early growth stage is a crucial component of earthquake early warning systems. Recent studies have revealed that the onsets of small and large earthquakes are variable but statistically indistinguishable. However, it is unknown whether small and large earthquakes can share the same processes at the same location. Here we show clear evidence of almost identical growth processes shared by repeating earthquakes of various sizes that have occurred in the Naka region, eastern Japan. Our results indicate that a large earthquake is a failure with a large characteristic spatial scale that is initially triggered by a failure with a small characteristic scale, which may also occur independently controlled by subtle differences in the physical conditions, suggesting the existence of a hierarchical structure on the plate interface. Earthquakes are random, but they may also be controlled by such structures.

[1] Department of Earth and Planetary Science, University of Tokyo, 7-3-1, Hongo, Bunkyoku, Tokyo 1113-0033, Japan. Correspondence and requests for materials should be addressed to T.O. (email: okuda.eps@gmail.com)

Do small and large earthquakes undergo different nucleation mechanisms? While capturing their initial rupture processes is a non-trivial task, these details are crucial in predicting the event sizes and improving earthquake early warning systems. Inference on the early evolution of a given earthquake lies between the two end-members of the conceptual earthquake rupture model: the nucleation model and cascade model[1–8]. The nucleation model, which is proposed primarily from observations in laboratory experiments[2,3], requires a precursory slow slip that is related to the final earthquake size. The observation of foreshocks is considered a by-product of the slow nucleation process in this model[9,10]. However, various types of scale independence[11,12] and the power-law nature[13] of natural earthquakes also suggest a large degree of self-similarity, and this forms the basis of the cascade model, in which the earthquake rupture grows as a cascading failure of hierarchical characteristic structures over a range of scales and without any distinguishable nucleation process[4].

The applicability of these two end-member models has been investigated by direct observations of the very beginning of seismic waveforms from earthquakes of various sizes. After more than two decades of debate[1,7,14–20], the conclusion seems to be that nucleation and rupture growth are statistically common for all earthquakes up to a magnitude limit that is determined by the tectonic environment[8]. Although this observation favours the cascade model, the applicability of the model, especially the existence of a hierarchical characteristic structure, is still unclear, because previous investigations of the initial nucleation of rupture in seismic waveforms employed either many seismograms from a broad spatiotemporal distribution of earthquakes, or just a few records from a series of closely located events. A more detailed analysis that employs many station records from a series of closely located earthquakes is therefore required to confirm the existence of a hierarchical characteristic structure.

An ideal target to investigate this potential hierarchical rupture growth is a group of earthquakes that occurred in the Naka region, off the coast of Ibaraki, eastern Japan, where the Pacific plate is subducting at a rate of 8.5 cm per year (Fig. 1). Hi-net, the high-sensitivity seismic observation network operated by the National Research Institute for Earth Science and Disaster Resilience (NIED) of Japan, has been in operation across the region since 2002. A repeating $M \sim 4.8$ earthquake sequence has been observed since 2003, consisting of five events to date. The recurrence interval was about 4 years for the first three events, and showed a slight decrease after the 2011 Tohoku-Oki earthquake ($M9.0$), probably due to the afterslip of the Tohoku-Oki earthquake (Fig. 1). A total of 68 earthquakes larger than $M1.5$ have occurred in this study region since 2002. The focal mechanisms determined by NIED for earthquakes larger than $M3.5$ are indicative of low-angle reverse faults, thus highlighting that these events occurred on the plate boundary.

In this study, we report clear evidence of almost identical growth processes shared by repeating earthquakes of various sizes in the Naka region. Raw seismograms from many stations show minimal differences in the very beginning of M5 earthquake waveforms, as well as those of smaller earthquakes, with these smaller events tending to occur as foreshocks in a broad sense. Our results indicate that a large earthquake is a failure with a large characteristic spatial scale that is initially triggered by a failure with a small characteristic spatial scale, which may also occur as an independent earthquake controlled by subtle differences in the physical conditions of the rupture process, thus

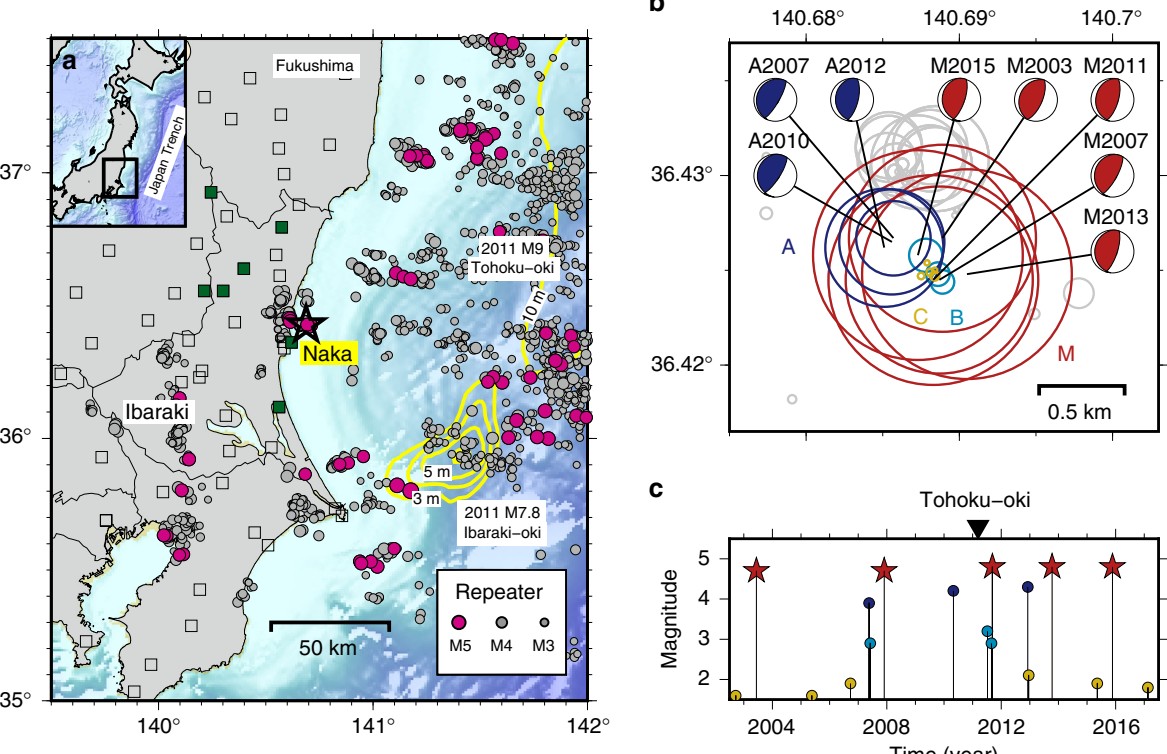

**Fig. 1** Seismicity of the Naka region, Japan. **a** Map showing the epicentres of the repeating earthquakes[26], with magenta and grey circles indicating $M > 4.5$ and smaller events, respectively. The yellow lines show the coseismic slip of the Tohoku-Oki earthquake[28] and the Ibaraki-Oki earthquake[29]. The green and open squares show the seismic stations used for seismogram comparison and relocation analysis, respectively. **b** Distributions of the Naka earthquakes. The centre and size of each circle represent the centroid location and expected source size, respectively. The red, blue, cyan, and yellow circles indicate groups M, A, B, and C, respectively. **c** Magnitude–time diagram of the grouped events. The triangle indicates the Tohoku-Oki earthquake

suggesting the existence of a time-independent hierarchical structure on the plate interface.

## Result

**Earthquake relocation**. We determined the relative location of the source centroid for 61 of these events using the relative arrival time differences measured by waveform cross-correlation[21]. The data consist of velocity seismograms recorded at 72 stations of Hi-net, the Japan Metrological Agency, The University of Tokyo, and Tohoku University, as summarised in Supplementary Table 1. Figure 1b shows the earthquake (circles) locations, with the circle sizes representing the expected source sizes for the events, calculated from their magnitudes and a stress drop of 38 MPa, which is estimated for another repeating earthquake sequence[22,23]. The estimated errors in the source locations vary and are dependent on the individual earthquakes, but the average standard deviations are ~50 m, ~100 m, and ~100 m in the north–south, east–west, and vertical directions, respectively.

Some of these earthquakes have been recognised as repeating earthquakes that rupture almost the same area each time[24]. Based on the repeating earthquake catalogue[24], the magnitude of the events, and the relocation results, we divided the earthquakes into four groups (M, A, B, and C) that represent the ~M4.8, ~M4,

~M3, and ~M2 events, respectively. Given the relative locations, expected source sizes, and possible errors, it is likely that the rupture areas of the group M events overlap with each other, as for groups A, B, and C. Furthermore, the rupture areas of the group M events include the rupture areas of the events in groups A, B, and C, which suggest a hierarchical structure that is embedded on the plate interface.

**Waveform comparisons of earthquakes of various sizes**. Some of these closely located earthquakes share quite similar initial waveforms. Hereafter, for the investigation of initial waveforms, we return to look at the original seismograms without any filtering. Figure 2 shows the raw data for M2015 (group M earthquake that occurred in 2015) and three group B events (B2007, B2011a, and B2011b). For each station, the waveforms are aligned at a common time delay for every event from their hypocentral times, to preserve relative arrival time differences at each station. The seismograms are neither filtered nor normalised. The raw data are ground velocity, which is almost proportional to the moment acceleration at the source. The almost identical P-wave onsets indicate that the rupture of M2015 and the group B events originated from approximately the same location. Their waveforms are also indistinguishable during the first 0.07 s of the

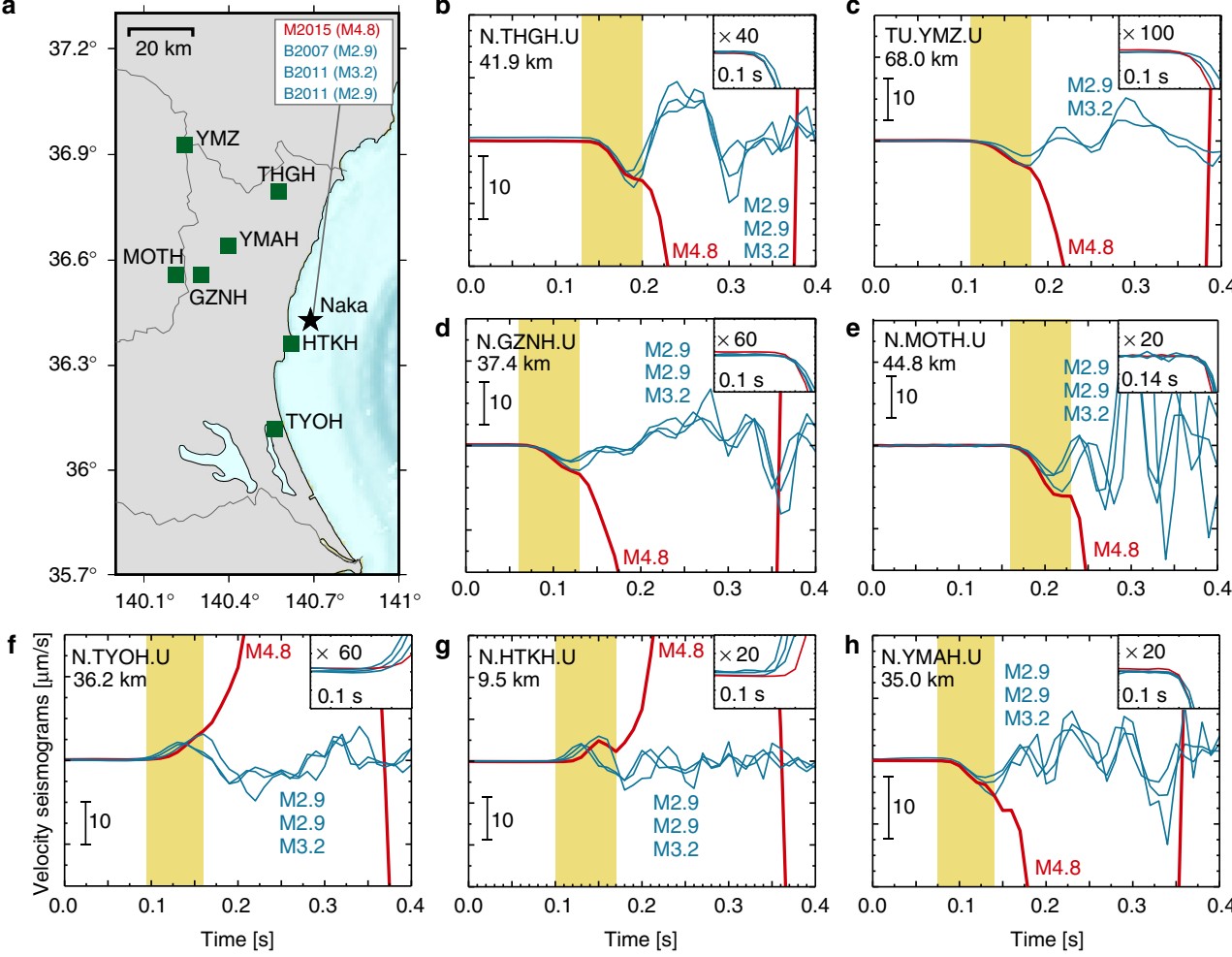

**Fig. 2** Seismogram comparison between the M2015 (M4.8) earthquake and three smaller group B earthquakes (M3.2, M2.9, and M2.9). **a** Map showing the locations of stations (squares) used in the waveform comparison, relative to the Naka events (star). Waveform comparison at each station: **b** N.THGH. **c** TU.TMZ. **d** N.GZNH. **e** N.MOTH. **f** N.TYOH. **g** N.HTKH. **h** N.YMAH. The vertical components of the raw seismograms are shown, with the onset of the P-waves magnified in the inset. The yellow zone indicates the 0.07 s time interval, which is the typical rupture duration of an M3 event, starting from the onset of P-waves. The relative arrival time differences at each station are preserved. The number under each station name shows the epicentre distance

selected events, which is comparable to the typical rupture duration of a *M*3 earthquake. Furthermore, M2015 nearly stopped growing at ~0.07 s, i.e., at the same time that marks the end of the rupture growth of the smaller events. Although, the Fig. 2 is made with seven stations, not to make a complicated figure, about 20 stations show the same feature.

We measured the similarity of waveforms using the mean of cross correlation (MCC) between seismograms from 0.1 s before to 0.07 s after the onset, calculated for every station. The stations used here for MCC calculation were the seven stations shown in Fig. 2. MCC values between M2015 and three group B events are 0.48, 0.72, and, 0.97, which are as high as the values among group

B events range 0.58–0.93. A slight reduction of MCC is responsible to the slight arrival-time delays at the southern stations that indicate a separation of ~100 m between M2015 and selected group B events (*M* ~ 3). This small difference indicates that all of these processes occurred within the typical area of a *M*3 earthquake, which is ~150 m for a stress drop of 38 MPa.

Figure 3 also compares seismograms of the other group M events with smaller earthquakes of groups A/B. Some stations had not operated in the part of analysis period due to the operational problem or the disaster of 2011 Tohoku-Oki earthquake. We calculated MCC value up to 0.13 s, using the stations shown in Fig. 3. MCC values are 0.82 and 0.95 between M2013 and two

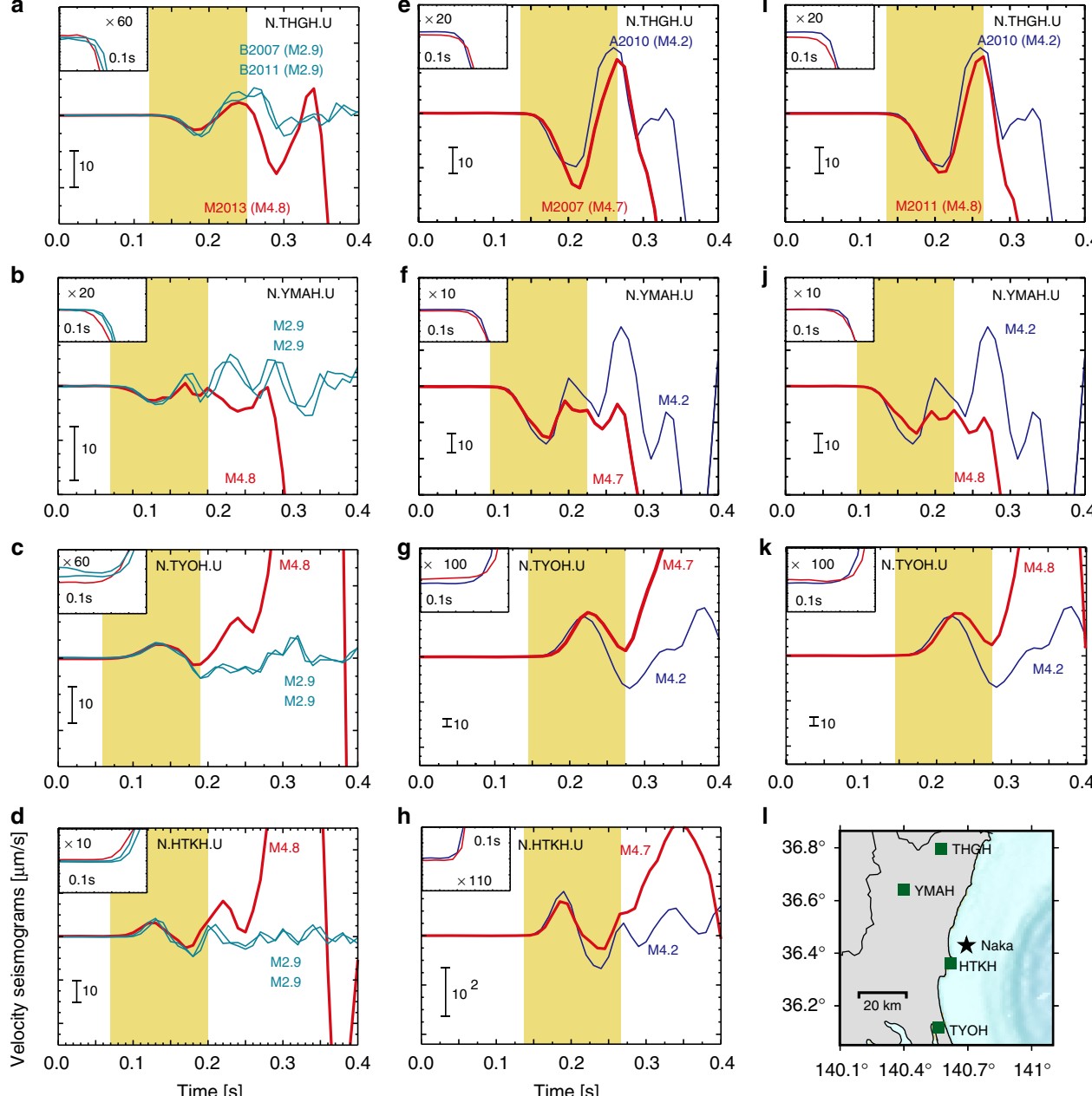

**Fig. 3** Seismogram comparison between the mainshocks and the smaller earthquakes. Waveform comparison between M2013 (*M*4.8), B2007 (*M*2.9), and B2011b (*M*2.9) at each station: **a** N.THGH. **b** N.YMAH. **c** N.TYOH. **d** N.HTKH. Waveform comparison between M2007 (*M*4.7) and A2010 (*M*4.2) at each station: **e** N.THGH. **f** N.YMAH. **g** N.TYOH. **h** N.HTKH. Waveform comparison between M2011 (*M*4.8) and A2010 (*M*4.2) at each station: **i** N.THGH. **j** N.YMAH. **k** N.TYOH. The vertical components of the seismograms are shown, with the onset of the P-waves magnified in the inset. The yellow zones mark the 0.13 s time interval where the larger and smaller events possess similar initial seismic responses. The relative arrival time differences at each station are preserved. **l** Map showing the locations of stations (squares) used in the waveform comparison, relative to the Naka events (star)

group B events (Fig. 3a, 4 stations), 0.88 between M2007 and A2010 (Fig. 3b, 4 stations), and 0.85 between M2011 and A2010 (Fig. 3c, 3 stations). These high MCCs support that the idea that these small earthquakes and moderate events started from almost the same location and show similar initial seismic waveforms. Yellow zones mark the 0.13 s time interval where the larger and smaller events possess similar initial seismic responses as reference. M2013, M2015, and the group B events started with the similar initial seismic responses, but the similarity of the M2013 and group B waveforms is observed for the first 0.13 s; whereas, the similarity with M2015 is only captured in the first 0.07 s.

However, at this moment, we cannot conclude that all group M events share common initial waveforms with some smaller events. Figure 4 shows the example of dissimilar waves, which compares seismograms of the other group M event, M2003, with smaller earthquakes of groups A/B. This is an event occurred during the construction of Hi-net and the number of stations are limited. Nevertheless, it is clear that no small earthquake shares the initial waveforms with the larger M2003 event. We can align the onsets of signal at one station, but the onsets at other stations do not coincide at all.

## Discussion

The behaviour of group M events is various since ~0.1 s after the onset. M2015 grew rapidly from 0.07 s, while M2007, M2011, and M2013 had a longer (~0.15 s) period with relatively small amplitude before their rapid growth (Fig. 3). In all cases except M2003, we find smaller events that share almost identical waveforms. Strong waveform similarity means that the rupture growth was highly similar in both space and time. This indicates that the details of the rupture growth can be repeatable to some extent, in that both the smaller and larger events grew similarly in the same area on the fault. Furthermore, it is also notable that the events with similar initial waveforms tend to occur far apart from each other in time. These imply that rupture growth is partly controlled by almost time-independent rupture conditions, such as local fault geometry.

It should be noted that the smaller earthquakes in groups A–C behave as foreshocks in a broad sense[25]: they occurred 12 times during the 15-year study period, with 8 of the 12 occurrences taking place in the year before a group M event (Fig. 1c). Assuming a binomial process, the probability of observing such a foreshock-like seismicity is <1.5%. Foreshocks generally occur within a few hours to days of the mainshock, and are considered

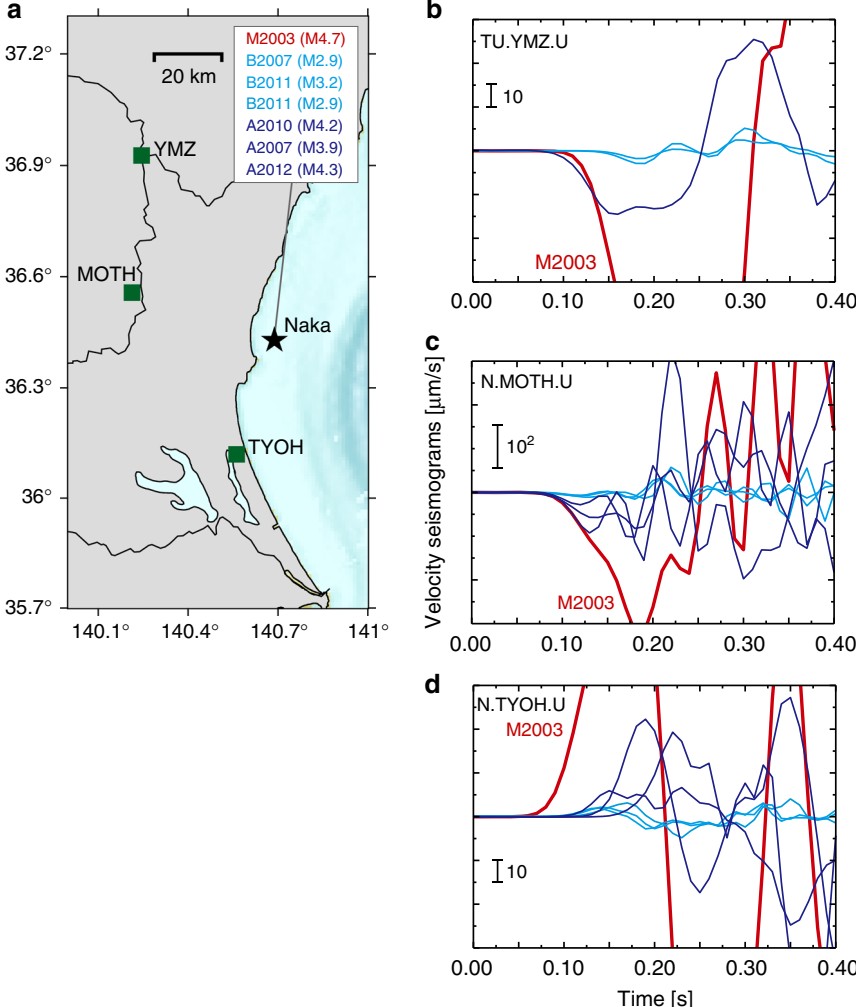

**Fig. 4** Seismogram comparison between the M2003 (M4.7) earthquake and smaller group A earthquakes. **a** Map showing the locations of stations (squares) used in the waveform comparison, relative to the Naka events (star). Waveform comparison at each station: **b** TU.YMZ. **c** N.MOTH. **d** N.TYOH. The vertical components of the raw seismograms are shown here. The traces are plotted to align the onsets of the P-waves. The seismograms are neither filtered nor normalised. The relative arrival time differences at each station are preserved

by-products of the mainshock nucleation[10,11,16]. However, foreshocks that occur within a few months to years of the mainshock indicate partial stress release in the mainshock area[26] or an increase in external stress levels[4]. Such foreshocks might have grown into mainshocks, but their growth stopped and they remained small.

It would be also important to note that the waveform of mainshock and its foreshocks tends to be not similar. Rupture starting from a small scale structure can grow into either small or large earthquakes. However, this feature indicates that once a small earthquake occurs, a large earthquake cannot start from exactly the same place, before the structure is sufficiently loaded again. Therefore, this small earthquake is considered as a foreshock of the large earthquake with dissimilar initial waveforms.

These examples indicate that when a small rupture occurs, it may stop as a small earthquake or it may grow into a large earthquake, depending on either the physical conditions of the fault interface or random factors. Figure 5 shows these processes using a patchy hierarchical structure. A hierarchical patch model is proposed here, in which a large strong patch contains smaller weak patches, with rupture tending to nucleate on these smaller patches. The failure of a small patch may trigger the failure of a larger patch, which is sometimes called the "cascade-up" process in numerical simulations[6]. A large earthquake in this model is the result of successive cascade-up processes from tiny to giant patches. The difference in the coloured durations in Figs. 2 and 3a suggests that the time required for a single cascade-up process may vary, as inferred from numerical simulations[6].

It is therefore important to estimate the probability of rupture from a smaller patch growing into the rupture of a larger patch, although our sample domain is currently limited to only five larger earthquakes. Our results indicate that the M2013 and M2015 events started from the rupture of the same small patch, which also occurred as independent group B earthquakes ($\sim$M3). Thus, the cascade-up probability from the rupture of the small patch ($\sim$M3) to the rupture of the mainshock patch (M4.8) can be estimated to be 40%.

In summary, the raw seismograms from many stations show minimal differences in the seismic waveforms during the first 0.07–0.13 s of the repeating $\sim$M5 Naka earthquake sequence, as well as those of smaller earthquakes, which tend to behave like foreshocks in a broad sense. This represents the first clear observations of such hierarchical rupture growth. Our results suggest the existence of a time-independent characteristic structure that is represented by a hierarchical patch distribution on the plate interface[4,9]. A large earthquake is the failure of a large patch, which is the consequence of successive cascade-up processes from a small patch. The small patch may also host an independent smaller earthquake. Whether a small patch rupture grows into a large-scale rupture or not depends on subtle differences in the physical conditions, which we cannot observe here, but may be able to evaluate in a probabilistic sense. Earthquakes are random processes, but they may not be completely random, as the pre-existing characteristics of the seismic region may control the rupture process to an extent. Evaluating the cascade-up probabilities would thus improve the probabilistic forecasting of earthquakes.

## Method

**Relocation method**. We relocated events using relative arrival times determined by waveform cross-correlation. This method consists of two main steps. First, we estimated the relative hypocentre location for each pair of events by maximising the summation of the waveform cross-correlation for all stations and components. We then determined the set of centroids that was consistent with the relative locations by solving a least-squares problem. This approach is described in ref. [24], with further information available in ref. [26]. We used a 4-s window, starting 1 s before the onset of the P-wave and shear-wave arrivals. A 1–10 Hz band-pass filter

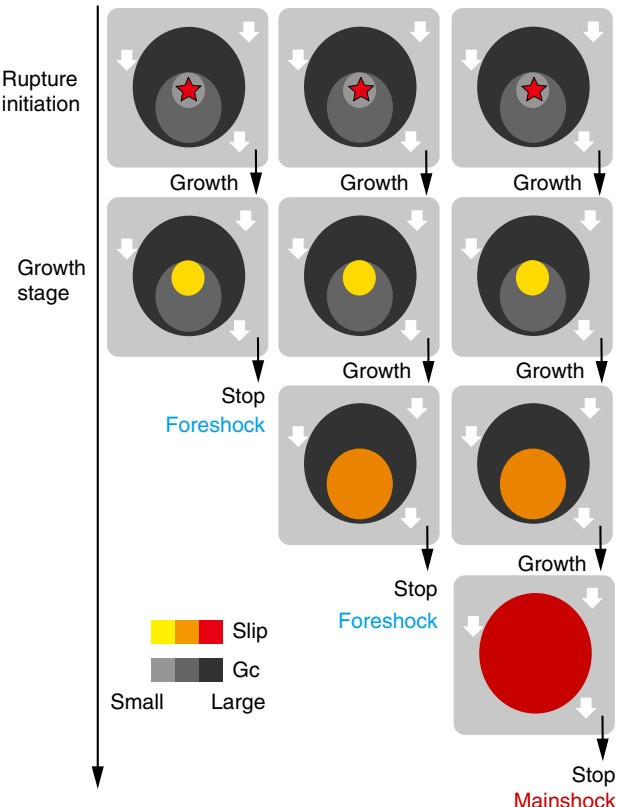

**Fig. 5** Conceptual hierarchical patch model. The hierarchical structure, defined by the different patch sizes in the figure, controls the overall earthquake behaviours. A large earthquake can be interpreted as the result of a cascade-up process that initiated from a small-patch rupture

was applied to the data prior to analysis. The seismic stations that acquired that data analysed here are equipped with 1-Hz 3-component seismometers, and the velocity seismograms are recorded at a sampling rate of 100 Hz.

**Pre-process of waveform comparisons**. For the seismogram comparison in Figs. 2, 3, and 4, we used raw velocity waveforms. The relative arrival time differences at each station are preserved, and the seismograms are neither filtered nor normalised. We subtracted the mean amplitude of the seismograms and this procedure affected the initial amplitude value. Our data in the waveform comparisons are basically raw data except for this mean subtraction.

## Data availability

All data from Hi-net, the Japan Metrological Agency, The University of Tokyo, and Tohoku University are available from http://www.hinet.bosai.go.jp/.

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

## Acknowledgements

We thank T. Nishikawa and D. Sato for discussions, and N. Uchida for providing the repeating earthquake catalog. Comments from the reviewers helped to improve the manuscript. This study was supported by the Ministry of Education, Culture, Sports, Science and Technology (MEXT) of Japan, under its Earthquake and Volcano Hazards Observation and Research Program, JSPS KAKENHI 16H02219, and MEXT KAKENHI 16H06477. All data were obtained from the NIED Hi-net data server. All maps in the figures were prepared using the Generic Mapping Tools[27].

## Author contributions

S.I. designed the study and constructed a system for the relocation analysis, and T.O. analysed the data.

## Additional information

**Competing interests:** The authors declare no competing interests.

