## [Peer Review File · Nature Communications]

Reviewers' comments:

Reviewer #1 (Remarks to the Author):

In the present work, the authors analyse the very beginning of raw seismograms, for a set of repeating earthquakes of various sizes that have occurred in the Naka region, eastern Japan. They show that the raw seismograms of M5 and smaller earthquakes from different stations show minimal differences in the first 0.07–0.13 s. According to the authors, their observations support the idea of almost identical growth processes and suggest the existence of a time-independent hierarchical structure to explain the differences in the rupture process leading to a small and a large earthquake.

Major comments:

1) the major concern regarding the work is that all the conclusions and interpretations are based on a qualitative, visual comparison of the recorded waveforms, which is generally subjective and not easy to justify.

For example, for Figure 2 (the same for Figure 3), the authors say "The almost identical P-wave onsets indicate that the rupture of M2015 and the group B events originated from approximately the same location. Their waveforms are also indistinguishable during the first 0.07 s of the selected events, which corresponds to the typical rupture duration of a M3 earthquake."

These figures are misleading, since data are shown in a time scale much longer than the portion used for the comparison (yellow bar). To my understandings and interpretation, the waveforms are not identical in the selected time window (yellow bar). Indeed, for some events, the initial amplitude reaches the maximum and then decreases in the considered time window. For other events, the waveform keeps increasing (or decreasing). To me, this makes a huge difference. Generally, it is not clear to me which is the parameters that the authors are using to quantitative characterize the initial similarity/difference in the waveforms. The formal definition of a quantitative parameter is missing and I think would be a strong point to support such a broad conclusion.

2) Data filtering: "A 1–10 Hz band-pass filter was applied to the data prior to analysis. The seismic stations that acquired that data analysed here are equipped with 1-Hz 3-component seismometers, and the velocity seismograms are recorded at a sampling rate of 100 Hz."

There is an intrinsic double limitation on data due to both the filtering procedure, which cancels all frequencies out of the range 1-10 Hz, and to the natural frequency of the instruments, which cancels all the frequencies lower than 1Hz. The magnitude range considered here goes from 1.5 to about 5. Assuming a stress drop value of 38MPa (as declared by the authors, which is however about 10 times larger than the expected value from worldwide earthquake, 3MPa), for a magnitude 2 earthquake, the expected corner frequency of the P-wave is about 10-30 Hz, which is well beyond the band limit of the applied filter. Similarly, for a magnitude 4 earthquake, with a stress drop of 38MPa, the expected corner frequency of the P-wave is about 2-3 Hz, which is again at the lower limit of the applied filter band. A magnitude 5 earthquake, has an expected corner frequency of about 1Hz (or smaller) and this would be completely hidden by both filter and instrumental response.

The effect of filtering is crucial. Indeed, when the cut-off frequency of the applied filter is close to the corner frequency of the earthquakes, the contribution of the source is removed and only the propagation effect will survive. Specifically, the band-pass filter applied here removes at the same time the lower frequencies of the largest events and the higher frequencies of the smallest earthquakes. I wonder whether the observed similarities of the waveforms could be due to the filtering and or to the instruments used. Please, comment.

3) I understand that the location techniques provides reliable location estimates, with pretty small uncertainties, of the order of 100 m maximum. However, even a small error on location (about 100 m), gives a uncertainty on the arrival time of the P-wave of about 0.015s, which is about 20% of the considered time window (0.07 s). Considering the very short time window used for the analysis, I think that the uncertainties might affect the results. Please, include a comment about that.

4) All the conclusions and interpretations are based on a limited number of earthquakes belonging to the same sequence, recorded at a few stations. I understand that 60 earthquakes recorded at 72 stations are analysed, but only few events at few stations are shown and discussed in the paper. Given the limited data set and since a quantitative analysis is missing (see previous comment), the observed features could therefore be a specific characteristic of the considered events / stations.

Minor comments:

1) Figure 1: the figure is misleading, since it gives the idea that a large number of events and recording stations are used for the analysis. Indeed, only the Naka region events are used and only few stations are shown in the paper. It would be better to show only the events which are effectively used in the work.

2) It is not clear to me how the events are associated into different groups and how the association how is used for the analysis.

3) I don't really understand if the band-pass filter is used only for the location of for the entire analysis (see Major comment n. 2)

4) Why does the initial amplitude before to the P-wave arrival is not zero? is there any data processing affecting the initial amplitude value? please, explain.

5) Figure 2 and 3 only show the behaviour of waveforms for some selected cases. What happens at the other stations? what is the behaviour of the observed shape as a function of distance? It would be useful to show some examples of waveforms also for larger distances.

6) Why does the considered time window changes to 0.13 (it was 0.07s in the first example)? how do the authors select the time window in which the signal is considered to be similar? Again, this is to show that the analysis are based on qualitative and subjective parameters, see Major comment n. 1.

7) I do not understand why the velocity signals are used for the analysis. I would expect that displacement signals are related to the source process (displacement is indeed a proxy for the Moment Rate Function), while velocity represents the time derivative of the Moment Rate Function.

Reviewer #2 (Remarks to the Author):

Dear Editors and Authors,

It was a pleasure to read and study this manuscript that takes a simple and elegant approach to the long-standing question of whether or not earthquakes are deterministic in their evolution. The results are quite spectacular, and - owing to the simplicity of the employed method - very convincing. I recommend the manuscript be published in Nature Communications after some moderate improvements to the text.

As the authors point out, it has been argued over for decades from what point on the final size of an ongoing rupture is determined. If the final size could be determined soon after the rupture onset (deterministic behavior) this would have some major impact on our understanding of dynamic rupture propagation, and would allow earthquake early warning (EEW) systems to provide longer warning times. While this would certainly be desirable, there is much observational evidence against such deterministic rupture behavior. From my personal perspective, this controversy has to a large extent been driven by observational studies that used highly processed data, and that have not been careful enough to rule out potential processing artifacts. In this study, on the other hand, the authors use unprocessed waveform data directly, which greatly simplifies the interpretation. They are able to do that by focusing on an extraordinarily well-recorded earthquake sequence in the Naka region.

What Okuda and Ide report is that they observe the same fault region to produce events that are highly similar in both their temporal and their spatial evolution. While some of these events terminate soon after their beginning and become small $M \sim 3$ events, in other cases their near-identical "replicas" keep growing and sometimes accelerate to become much bigger events ($M \sim 4.8$). The fact that the smaller and the larger events have the same initial evolution suggests that their evolution is affected, if not governed, by a non-transient condition, likely fault structure.

These observations, and their implications, are rather unprecedented in their clarity and will make it difficult for others to argue for a strongly deterministic rupture behavior. The discussion on earthquake determinism has always been of prime importance for earthquake source physics, but it has now gained additional importance with the feasibility of earthquake early warning systems: if EEW systems are designed based on a false expectation of deterministic rupture behavior, at least part of their potential to provide useful alerts will be squandered.

While I have very few comments on the manuscript content, I have a few suggestions that could improve the readability of the manuscript. Although it is generally very well written and easy to read there are some descriptions that are in their present form rather vague. They could gain from a re-formulation, with more specific descriptions. I have made specific suggestions in the comments listed below.

Great work, I am looking forward to reading the final version of the article!

Best regards,
Men-Andrin Meier

MAIN

The comments below only concern how the material is presented, not the material itself. The main comment is that some of your formulations are rather vague and should be made more specific.

- Here is a, fully optional, suggestion for how I would re-formulate your main observations, including a comparison of the largest events among themselves:
 - the M_{2015} event has a slight slow-down at $\sim 0.07s$, after which it accelerates rapidly and becomes a $M_{4.8}$ event. The other 3 $M \sim 5$ events have a somewhat longer period of slow growth rates, before they grow rapidly at $\sim 0.2s$. But for all cases, there are smaller events that share almost identical waveform shapes. The waveform similarity remains strong for almost the entire rupture duration of the small events, and includes small details in the rupture evolution, such as the slow-down of M_{2015} after $0.07s$. Such strong waveform similarity can only be obtained if the rupture evolution was highly similar in both space and time. This makes the details of the rupture evolution to some extent repeatable, in that both the smaller and larger events slowed down and accelerated in the same area on the fault. Which implies that rupture evolution is at least partly governed by long-term rupture conditions (e.g. the details of local fault surface topography and of the stress field).
 - I think you are basically saying that very same thing, but I have here tried here to suggest a more explicit formulation.
- You should also mention why velocity records directly inform about the moment rate acceleration.

- You do not much discuss the time difference between the events you compare. Are the ones that are farther in time more different in shape or not? If I understand correctly the events with similar waveforms can be far apart from each other in time. This is what suggests that the rupture evolution is affected not by transient random effect, but by a longer-term condition, such as fault structures, e.g. rupture barriers that often, but not always, stop ruptures in the same place. Therefore, your argument would be strengthened if you explicitly state that not just the immediate foreshocks are similar.
- I would more clearly separate the direct implications of the observations and the proposed hierarchical model. The former is that the studied fault region repeatedly produces event sequences where small and large events have near-identical evolution for some limited time period. The hierarchical model is consistent with these observations, but it is not directly required by them, in that a non-hierarchical cascade model could lead to similar observations.

FIGURES

- Figure 1: I would somehow highlight the Naka cluster, since that's what you study in this paper.
- Not sure how/why you selected the records that you compare in Figures 2 & 3. This should be clearly stated. E.g. the top right subfigure in Figure 2 is the only one that only shows 2 of the M₋₃ events. Is this because the data is unavailable from the third event, or is the waveform different for this station? In general, I would assume there are also records that have different onset shapes. This would suggest that there can be different source evolutions in very close spatial proximity. If that's the case I would describe that fact explicitly in the text.
- In Figures 2 & 3 it is a bit difficult to figure out what you're comparing against what, partly because, as stated above, the selections are not clearly motivated. Maybe you could try to add a small overview panel, or just add some details of the events that are being compared in the station map.
- You could add the recording distance (e.g. with respect to the largest event) to the station name in each panel.

DETAILS

- Abstract: "characteristic scale" is a "characteristic spatial scale" (?)
- the M, A, B, C naming for the event groups is non-informative, and somewhat confusing, e.g. in Figure 3 where a, b, and c are also used to indicate figure parts. Why not just name them M₅, M₄, M₃ and M₂ groups? Or, for the M_{4.8} events you could use something like M_{MS}, for "main shocks".
- Line 102: "relative arrival time differences are preserved..." is unclear. I think you are saying that the waveforms are plotted at their actual time relative to the origin time, rather than using the time of the best waveform cross-correlation fit. Same in caption to Figure 3.
- Line 108: "tended to be flat"; I would reformulate along the lines of: "M₂₀₁₅ temporarily stops growing/accelerating at $\sim 0.07s$, i.e. at the same time that marks the end of the rupture acceleration of the smaller events." The fact that this bump happens at the same time for the smaller and larger events is truly impressive and should be stated clearly.
- Line 153: use a colon to point out the direct connection between the two statements, as in "in a broad sense²⁸: they occurred ..."
- Line 164: skip the "For example"
- Figure 4 had no caption in my file, but that's probably a processing error.

Response to Reviewers' comments:

Thank you for the review of our manuscript entitled "Hierarchical rupture growth evidenced by the initial seismic waveforms". We appreciate your positive evaluation and helpful comments by reviewers. Here we provide our response to comments by the reviewers. The main points of the revision are:

- Additional explanations about data handling including filtering.
- Quantitative evaluation of waveform similarity.

We also made many small improvements as suggested by reviewers.

In the following pages, we provide one to one replies for the reviewer comments, which are shown by red characters. Please refer to the improved manuscript for detailed revisions. Line numbers mentioned in this reply are of the manuscript without change history.

Best regards,
Takashi Okuda

Reviewer #1

Major comments:

1) the major concern regarding the work is that all the conclusions and interpretations are based on a qualitative, visual comparison of the recorded waveforms, which is generally subjective and not easy to justify. For example, for Figure 2 (the same for Figure 3), the authors say "The almost identical P-wave onsets indicate that the rupture of M2015 and the group B events originated from approximately the same location. Their waveforms are also indistinguishable during the first 0.07 s of the selected events, which corresponds to the typical rupture duration of a M3 earthquake." These figures are misleading, since data are shown in a time scale much longer than the portion used for the comparison (yellow bar). To my understandings and interpretation, the waveforms are not identical in the selected time window (yellow bar). Indeed, for some events, the initial amplitude reaches the maximum and then decreases in the considered time window. For other events, the waveform keeps increasing (or decreasing). To me, this makes a huge difference. Generally, it is not clear to me which is the parameters that the authors are using to quantitative characterize the initial similarity/difference in the waveforms. The formal definition of a quantitative parameter is missing and I think would be a strong point to support such a broad conclusion.

Thank you for thoughtful comments. We have added a quantitative analysis to evaluate the waveform similarity objectively. We measured the similarity of waveforms using the mean of cross correlation (MCC) between seismograms from 1 s before to 0.07 s after the onset, calculated for every station. For example, in the case of the events shown in Figure 2, we calculated MCC values using the 7 stations shown in the figure. MCC values between M2015 and three group B events are 0.55, 0.75, and, 0.98, which are as high as the values among group B events range 0.63-0.94. These high MCCs support that the idea that these small earthquakes and moderate events started from almost the same location and show similar initial seismic waveforms.

2) Data filtering: "A 1–10 Hz band-pass filter was applied to the data prior to analysis. The seismic stations that acquired that data analysed here are equipped with 1-Hz 3-component seismometers, and the velocity seismograms are recorded at a sampling

rate of 100 Hz." There is an intrinsic double limitation on data due to both the filtering procedure, which cancels all frequencies out of the range 1-10 Hz, and to the natural frequency of the instruments, which cancels all the frequencies lower than 1Hz. The magnitude range considered here goes from 1.5 to about 5. Assuming a stress drop value of 38MPa (as declared by the authors, which is however about 10 times larger than the expected value from worldwide earthquake, 3MPa), for a magnitude 2 earthquake, the expected corner frequency of the P-wave is about 10-30 Hz, which is well beyond the band limit of the applied filter. Similarly, for a magnitude 4 earthquake, with a stress drop of 38MPa, the expected corner frequency of the P-wave is about 2-3 Hz, which is again at the lower limit of the applied filter band. A magnitude 5 earthquake, has an expected corner frequency of about 1Hz (or smaller) and this would be completely hidden by both filter and instrumental response. The effect of filtering is crucial. Indeed, when the cut-off frequency of the applied filter is close to the corner frequency of the earthquakes, the contribution of the source is removed and only the propagation effect will survive. Specifically, the band-pass filter applied here removes at the same time the lower frequencies of the largest events and the higher frequencies of the smallest earthquakes. I wonder whether the observed similarities of the waveforms could be due to the filtering and or to the instruments used. Please, comment.

We suspect that the reviewer made a misunderstanding on our data handling, and the reason might be in our way of explanation in the previous manuscript. As another reviewer correctly evaluated, the most important point of this paper is the comparison of raw data without processing. No band-pass filter was used in the waveform comparison (Figs. 2 and 3), as explicitly written in the original manuscript. However, a bandpass filter of 1-10 Hz was used for relocation analysis, which might be misleading for some readers. We have clarified this point in the manuscript by adding a sentence: "Hereafter, for the investigation of initial waveforms, we return to look at the original seismograms without any filtering." (L105) We also added a paragraph in Method section.

3) I understand that the location techniques provides reliable location estimates, with pretty small uncertainties, of the order of 100 m maximum. However, even a small error on location (about 100 m), gives a uncertainty on the arrival time of the P-wave of about

0.015s, which is about 20% of the considered time window (0.07 s). Considering the very short time window used for the analysis, I think that the uncertainties might affect the results. Please, include a comment about that.

This comment might come from another kind of misunderstanding. Our waveform comparison based on the direct observation is possible without any information about the hypocenter location. For example, we can compare all combinations of events in our catalog, but it is too inefficient. The comparison should be limited for events occurred closely to each other. Therefore, we demonstrated the proximity of hypocenter locations and identified the repeating earthquake clusters using the relocation analysis. There are relocation errors, of course, but they do not affect our conclusion at all. We admit the difference of relocation analysis and waveform comparison in the manuscript was not clear enough, and clarified this point in the revised one.

4) All the conclusions and interpretations are based on a limited number of earthquakes belonging to the same sequence, recorded at a few stations. I understand that 60 earthquakes recorded at 72 stations are analysed, but only few events at few stations are shown and discussed in the paper. Given the limited data set and since a quantitative analysis is missing (see previous comment), the observed features could therefore be a specific characteristic of the considered events / stations.

We partially agree with this comment. We are not arguing that such similarity of waveforms is always observed. Observed features might be specific for the considered events. However, the number of stations are not a few. Although the Fig. 2 is made with 7 stations, not to make a complicated figure, almost all available (about 20) stations show the same feature. We emphasized this point in the revised manuscript (L117). The example of dissimilar waves was also shown in Fig. S2, which we moved into the main body. Thus, we here report rare, or at least so far overlooked phenomenon, which is clear evidence that events of different size can share identical initial rupture process suggesting hierarchical nature of earthquake source.

Minor comments:

1) Figure 1: the figure is misleading, since it gives the idea that a large number of events and recording stations are used for the analysis. Indeed, only the Naka region events are used and only few stations are shown in the paper. It would be better to show only the events which are effectively used in the work.

We have modified the color of stations used in Figure 1.

2) It is not clear to me how the events are associated into different groups and how the association how is used for the analysis.

We made grouping based on the relocation result, the magnitude of events, and also the repeating earthquake catalogue of Uchida and Matsuzawa (2013), which searched the repeating earthquake ($M > 2.5$) distribution in Tohoku region (including Naka region) for long-term (27 year) by the waveform similarity analysis. We clarified it in the text. Each earthquake groups may represent a patchy like structure on the plate boundary as we discussed the hierarchical patch model in the main text.

3) I don't really understand if the band-pass filter is used only for the location of for the entire analysis (see Major comment n. 2)

Band-pass filter was used only for the relocation. We used raw seismograms for waveform comparisons. We clarified this point in the manuscript.

4) Why does the initial amplitude before to the P-wave arrival is not zero? is there any data processing affecting the initial amplitude value? please, explain.

We subtracted the mean amplitude of the seismograms and this procedure affected the initial amplitude value. Our data in the waveform comparisons are basically raw data except for this mean procedure. We added explanations about this process.

5) Figure 2 and 3 only show the behaviour of waveforms for some selected cases. What happens at the other stations? what is the behaviour of the observed shape as a function of distance? It would be useful to show some examples of waveforms also for larger

distances.

Waveform similarity was observed at both near-field and far-field stations. To highlight this point, we have added the information of epicenter distance in Figure 2. We also explained roughly how many stations observed similar waves in the text. We added an explanation figure just for reviewer to show waveforms at many stations for one combination of events.

20151122082026 36.429 140.688 52.1 4.8
20110710185450 36.426 140.697 51.6 3.2

20151122082026 36.429 140.688 52.1 4.8
20110710185450 36.426 140.697 51.6 3.2

Figure. Seismogram comparison between the M2015 ($M4.8$) earthquake and group B earthquake ($M3.2$). The vertical components of the raw velocity seismograms are shown in the right side. The vertical components of the acceleration seismograms are shown in the central and left side. The relative arrival time differences at each station are preserved.

6) Why does the considered time window changes to 0.13 (it was 0.07s in the first example)? how do the authors select the time window in which the signal is considered to be similar? Again, this is to show that the analysis are based on qualitative and subjective parameters, see Major comment n. 1.

We determined the window length based on earthquake magnitudes and similarity durations for each earthquake pairs. This length is a kind of reference, which need not be determined by rigorous objective measures, and does not have crucial importance. We clarified this point in the text. We understand that this duration (0.07 in the first example) could change even in the similar magnitude earthquake pairs occurred in the same place as we showed the seismogram comparisons of M2015-group B and M2013-group B. Theoretical studies of Noda et al. (2013, 2014) are documented in more detail, which showed that the time required for a single cascade-up process may vary on the hierarchical patches in the rate-and-state framework.

7) I do not understand why the velocity signals are used for the analysis. I would expect that displacement signals are related to the source process (displacement is indeed a proxy for the Moment Rate Function), while velocity represents the time derivative of the Moment Rate Function.

This is because raw data is the velocity signal. To convert the velocity signal into the displacement signal, a preprocess is necessary and it enlarges the low frequency noise. In the view point of simplicity, we emphasized using the raw data.

Dear Reviewer Men-Andrin Meier,

Here is a, fully optional, suggestion for how I would re-formulate your main observations, including a comparison of the largest events among themselves:

- o the M2015 event has a slight slow-down at ~ 0.07 s, after which it accelerates rapidly and becomes a M4.8 event. The other 3 M ~ 5 events have a somewhat longer period of slow growth rates, before they grow rapidly at ~ 0.2 s. But for all cases, there are smaller events that share almost identical waveform shapes. The waveform similarity remains strong for almost the entire rupture duration of the small events, and includes small details in the rupture evolution, such as the slowdown of M2015 after 0.07s. Such strong waveform similarity can only be obtained if the rupture evolution was highly similar in both space and time. This makes the details of the rupture evolution to some extent repeatable, in that both the smaller and larger events slowed down and accelerated in the same area on the fault. Which implies that rupture evolution is at least partly governed by long-term rupture conditions (e.g. the details of local fault surface topography and of the stress field).
- o I think you are basically saying that very same thing, but I have here tried here to suggest a more explicit formulation.

Thank you for giving me a lot of thoughtful and constructive comments. We have revised our manuscript based on your suggestions as listed below. We added following words;

“The behavior of group M events is various since ~ 0.1 s after the onset. M2015 grew rapidly from 0.07 s, while M2007, M2011, and M2013 had a longer (~ 0.15 s) period with relatively small amplitude before their rapid growth (Fig. 3). In all cases except M2003, we find smaller events that share almost identical waveforms. Strong waveform similarity means that the rupture growth was highly similar in both space and time. This indicates that the details of the rupture growth can be repeatable to some extent, in that both the smaller and larger events grew similarly in the same area on the fault. Furthermore, it is also notable that the events with similar initial waveforms tend to occur far apart from each other in time. These imply that rupture growth is partly

controlled by almost time-independent rupture conditions, such as local fault geometry.”

- You should also mention why velocity records directly inform about the moment rate acceleration.

We have clarified this point in the manuscript as following;

“The raw data is ground velocity, which is almost proportional to the moment acceleration at the source.”(L110).

- You do not much discuss the time difference between the events you compare. Are the ones that are farther in time more different in shape or not? If I understand correctly the events with similar waveforms can be far apart from each other in time. This is what suggests that the rupture evolution is affected not by transient random effect, but by a longer-term condition, such as fault structures, e.g. rupture barriers that often, but not always, stop ruptures in the same place. Therefore, your argument would be strengthened if you explicitly state that not just the immediate foreshocks are similar.

Thank you for providing interesting insight. We have added the discussion of time difference between the events we compare. Moreover, we added information about the event occurrence time in Figure 2 to help the understanding of the discussion.

- I would more clearly separate the direct implications of the observations and the proposed hierarchical model. The former is that the studied fault region repeatedly produces event sequences where small and large events have near-identical evolution for some limited time period. The hierarchical model is consistent with these observations, but it is not directly required by them, in that a non-hierarchical cascade model could lead to similar observations.

We changed the order of explanations, and moved the discussion on the hierarchical cascade model to the later part. As you mentioned, the hierarchical cascade model is just one of the reasonable interpretations.

FIGURES

- Figure 1: I would somehow highlight the Naka cluster, since that's what you study in this paper.

We modified the color of stations and Naka cluster in Figure 1.

- Not sure how/why you selected the records that you compare in Figures 2 & 3. This should be clearly stated. E.g. the top right subfigure in Figure 2 is the only one that only shows 2 of the M~3 events. Is this because the data is unavailable from the third event, or is the waveform different for this station? In general, I would assume there are also records that have different onset shapes. This would suggest that there can be different source evolutions in very close spatial proximity. If that's the case I would describe that fact explicitly in the text.

Yes, some data are unavailable in Fig. 2. Our analysis period is more than 15 years. Some stations had not operated in the part of analysis period. In the waveform comparisons, we tried not to be as arbitrary as possible. We clarified this point in the paper.

- In Figures 2 & 3 it is a bit difficult to figure out what you're comparing against what, partly because, as stated above, the selections are not clearly motivated. Maybe you could try to add a small overview panel, or just add some details of the events that are being compared in the station map.

Based on your comment, we have added information about the event occurrence time in figures 2 & 3 to clarify what we're comparing against what.

Moreover, we moved the figure of the other event waveforms into the main body from the supplementary materials to clarify the motivation of selection. This new figure could help to understand that the observed features might be a specific characteristic of the considered events. We here report very rare phenomenon, or at least so far overlooked phenomenon, which is clear evidence that events of different size can share identical initial rupture process suggesting hierarchical nature of earthquake source.

- You could add the recording distance (e.g. with respect to the largest event) to the station name in each panel.

We have added this point as such.

DETAILS

- Abstract: "characteristic scale" is a "characteristic spatial scale" (?)

We have revised this point as such.

- the M, A, B, C naming for the event groups is non-informative, and somewhat confusing, e.g. in Figure 3 where a, b, and c are also used to indicate figure parts. Why not just name them M5, M4, M3 and M2 groups? Or, for the M4.8 events you could use something like M_{MS} , for "main shocks".

Thank you for your suggestion. We have considered M5, M4, M3 and M2 naming, but the M, A, B, C naming is useful to describe the occurrence time like M2015. As you mentioned, Figure 3 is somewhat confusing. Therefore, we have simplified the Figure 3 by removing some group names.

- Line 102: "relative arrival time differences are preserved..." is unclear. I think you are saying that the waveforms are plotted at their actual time relative to the origin time, rather than using the time of the best waveform cross-correlation fit. Same in caption to Figure 3.

We agree that it was a bit unclear. We improved the expression.

- Line 108: "tended to be flat"; I would reformulate along the lines of: "M2015 temporarily stops growing/accelerating at ~ 0.07 s, i.e. at the same time that marks the end of the rupture acceleration of the smaller events." The fact that this bump happens at the same time for the smaller and larger events is truly impressive and should be stated clearly.

We have revised this point as such.

- Line 153: use a colon to point out the direct connection between the two statements, as in "in a broad sense²⁸: they occurred ..."

We have revised this point as such.

- Line 164: skip the "For example"

The word "For example" is removed.

- Figure 4 had no caption in my file, but that's probably a processing error.

Thank you for your information.

Reviewers' comments:

Reviewer #1 (Remarks to the Author):

>> In my opinion, the parameter adopted does not respond to the request of quantitatively proving the similarity between the waveforms. The MCC indeed is computed using 1.07 s time windows, containing 1 s of pre-event noise and 0.07 s of P-wave signal, so that the useful signal portion is less than 10% of the whole window. In this way, I believe that the MCC is mostly affected by the pre-event noise signal. I might be wrong, but I think that the same parameter should be measured on the P-wave signal only (or with a very small pre-event time window) to quantify the differences in the initial waveforms.

>> I think that the window length is crucial for the comparison of the signal and cannot be arbitrarily chosen. By absurd, if you would use a very short time window (i.e., the sampling of data) you could not appreciate any difference/similarity in the waveforms. Similarly, if you would use the whole signal, the waveform feature could not be attributed to the source effect, but the contribution of path propagation effect should be considered. The time window for comparing the waveforms is crucial, indeed.

For each magnitude earthquake, the rupture process has a different duration. The similarity/difference should be observed in a time window which is consistent with (comparable or a fraction of) the rupture process of the earthquake itself.

Reviewer #1

In my opinion, the parameter adopted does not respond to the request of quantitatively proving the similarity between the waveforms. The MCC indeed is computed using 1.07 s time windows, containing 1 s of pre-event noise and 0.07 s of P-wave signal, so that the useful signal portion is less than 10% of the whole window. In this way, I believe that the MCC is mostly affected by the pre-event noise signal. I might be wrong, but I think that the same parameter should be measured on the P-wave signal only (or with a very small pre-event time window) to quantify the differences in the initial waveforms. I think that the window length is crucial for the comparison of the signal and cannot be arbitrarily chosen. By absurd, if you would use a very short time window (i.e., the sampling of data) you could not appreciate any difference/similarity in the waveforms. Similarly, if you would use the whole signal, the waveform feature could not be attributed to the source effect, but the contribution of path propagation effect should be considered. The time window for comparing the waveforms is crucial, indeed. For each magnitude earthquake, the rupture process has a different duration. The similarity/difference should be observed in a time window which is consistent with (comparable or a fraction of) the rupture process of the earthquake itself.

Thank you for careful comments. We checked the stability of MCC values by changing the pre-signal time. In the revised text, we adopted the pre-signal time of 0.1 s, so that the useful signal portion is about 50% of the whole window. Actually, as shown below, MCC values do not depend much on the window setting, which ensures the stability of our results.

Previous manuscript (1.0 s pre-signal time):

- MCC values between M2015 and three group B events are 0.55, 0.75, and, 0.98, which are as high as the values among group B events range 0.63-0.94.
- MCC values are 0.82 and 0.91 between M2013 and two group B, 0.88 between M2007 and A2010, and 0.87 between M2011 and A2010.

Revised manuscript (0.1 s pre-signal time):

- MCC values between M2015 and three group B events are 0.48, 0.72, and, 0.97, which are as high as the values among group B events range 0.58-0.93.

• MCC values are 0.82 and 0.95 between M2013 and two group B, 0.88 between M2007 and A2010, and 0.85 between M2011 and A2010.